# Phenolic Compounds and Ginsenosides in Ginseng Shoots and Their Antioxidant and Anti-Inflammatory Capacities in LPS-Induced RAW264.7 Mouse Macrophages

**DOI:** 10.3390/ijms20122951

**Published:** 2019-06-17

**Authors:** Fan Yao, Qiang Xue, Ke Li, Xinxin Cao, Liwei Sun, Yujun Liu

**Affiliations:** 1College of Biological Sciences and Biotechnology, Beijing Forestry University, Qinghuadonglu No. 35, Haidian District, Beijing 100083, China; Fany_strive@163.com (F.Y.); xuetian20130607@163.com (Q.X.); like17931@163.com (K.L.); 15501080616@163.com (X.C.); lsun2013@bjfu.edu.cn (L.S.); 2Beijing Beilin Advanced Eco-Environmental Protection Technology Institute Co. Ltd., Qinghuadonglu No. 35, Haidian District, Beijing 100083, China

**Keywords:** *Panax ginseng* shoot extract (GSE), ginsenoside, phenolic compounds, antioxidant capacity, anti-inflammatory capacity, RAW264.7 mouse macrophages

## Abstract

We conducted this study for the first time to evaluate changes in the composition and contents of phenolic compounds and ginsenosides in ginseng shoot extracts (GSEs) prepared with different steaming times (2, 4, and 6 h) at 120 °C, as well as their antioxidant and anti-inflammatory activities in lipopolysaccharide (LPS)-induced RAW264.7 mouse macrophages (RAW264.7 cells). The results show that total phenol and flavonoid contents were both significantly higher in steamed versus raw GSEs, and the same trend was found for 2,2′-diphenyl-1-picrylhydrazyl (DPPH•) and 2,2′-azobis-(3-ethylbenzothiazoline-6-sulfonic acid) (ABTS•^+^) scavenging capacities. Among the 18 ginsenosides quantified using high-performance liquid chromatography (HPLC) with the aid of pure standards, polar ginsenosides were abundant in raw GSEs, whereas less-polar or rare ginsenosides appeared after steaming at 120 °C and increased with steaming time. Furthermore, steamed GSEs exhibited a greater ability to inhibit the production of inflammatory mediators and pro-inflammatory cytokines, such as nitric oxide (NO), interleukin (IL)-6, and tumor necrosis factor (TNF)-α in LPS-induced RAW264.7 cells at the same concentration. Relative expression levels of inducible nitric oxide synthase (iNOS), IL-6, TNF-α, and cyclooxygenase-2 (COX-2) mRNAs were attenuated by the GSEs, probably due to the enrichment of less-polar ginsenosides and enhanced antioxidant activity in steamed GSEs. These findings, combined with correlation analysis, showed that less-polar ginsenosides were major contributors to the inhibition of the overproduction of various inflammatory factors, while the inhibitory effects of total phenols and total flavonoids, and their antioxidant abilities, are also important.

## 1. Introduction

Inflammation, a defense reaction to alert the immune system and protect the host from infection with various pathogens such as bacteria, viruses, and fungi, is generally beneficial to the host due to its ability to repair injured tissues and maintain homeostasis, which are regulated by numerous inflammatory mediators and cytokines including nitric oxide (NO), interleukin-6 (IL-6), and tumor necrosis factor (TNF-α) [1]. When inflammation is not well managed, excessive inflammatory factors are produced, leading to chronic inflammatory diseases such as rheumatoid arthritis, multiple sclerosis, systemic lupus erythematosus, and type I diabetes [2]. Moreover, recurring inflammation may trigger tumors by stimulating oxidative stress, generating excess reactive oxygen species, accelerating cellular proliferation, and inducing DNA damage [3]. Therefore, inhibiting the overproduction of inflammatory factors is essential to forestalling the pathogenesis and progression of various disorders.

Ginseng (*Panax ginseng* C.A. Meyer), containing ginsenosides in the root as its main bioactive ingredients, is a perennial plant in the family Araliaceae that has long been used as a traditional herb in China, Korea, Japan, and Southeast Asia for treating various diseases [4,5,6,7,8,9,10]. Numerous studies have demonstrated that red ginseng—produced when fresh ginseng undergoes repeated steaming and drying—exhibits higher anticancer potential than fresh ginseng due to its abundant contents of less-polar or rare ginsenosides, such as ginsenosides Rg_3_, Rg_5_, Rk_1_ and Rh_2_, which are generated during steaming [11,12]. Red ginseng, which causes fewer side effects compared to fresh ginseng, not only exhibits protective effects against microbial infections and positive effects on memory [13,14], but also attenuates the pro-oxidant conditions associated with chronic diseases and improves peripheral circulation disorders [15,16]. Although red ginseng possesses various health-promoting effects, no study has compared the anti-inflammatory effects between red and white ginseng.

When considering the herb ginseng, most previous studies focused on its root, despite its stems, leaves, flowers, and berries are also good sources of ginsenosides [17,18,19], and qualitative and quantitative analyses showing that ginsenosides vary greatly among parts of the ginseng plant [20]. Reports have also shown that ginsenosides, polyphenols, and the antioxidant activity of hydroponic-cultured ginseng roots changed simultaneously with increased heating temperature, and that the antioxidant activity of hydroponic-cultured ginseng leaf was greater than that of the root [21]. Unfortunately, no studies have focused on the anti-inflammatory effects of ginseng leaves or shoots extracts. In the present study, we examined the contents of total phenols and total flavonoids, and the 2,2′-diphenyl-1-picrylhydrazyl (DPPH) and 2,2′-azobis-(3-ethylbenzothiazoline-6-sulfonic acid) (ABTS) antioxidant abilities of the extracts from ginseng shoots steamed at 120 °C for different times for the first time; clarified effects of steaming on transformation mechanisms from polar to less-polar ginsenosides in ginseng shoots; and systematically and comprehensively compared the anti-inflammatory effects of steamed and un-steamed ginseng shoots extracts (GESs) in lipopolysaccharide (LPS)-induced RAW264.7 mouse macrophages (RAW264.7 cells), along with identification of the composition and determination of the contents of ginsenosides of the extracts. Through this study, a concise method for utilizing the natural products in ginseng shoots was established, laying the foundation for further development of ginseng shoot resources.

## 2. Results and Discussion

### 2.1. Total Phenols, Total Flavonoids, and Antioxidant Activities in GSEs

Total phenols in GSEs exhibited a significant increase with steaming time from 0 to 6 h (Figure 1A), and a similar increasing trend in total flavonoids was also observed, but with no significant difference among GSE2, GSE4 and GSE6 (Figure 1B). The effects of steaming on antioxidant activities were subsequently estimated (Figure 1C). DPPH• scavenging activity rose significantly (*p* < 0.05) with steaming time from 0 to 4 h, and then decreased slightly at 6 h, with no significant difference (*p* < 0.05) from that at 4 h. On the other hand, ABTS•^+^ scavenging activity was enhanced with steaming time from 0 through 6 h, with no significant differences among steaming times longer than 2 h (Figure 1D).

Polyphenols are known as safe, natural antioxidants that are capable of preventing several chronic diseases including heart disease, cancer, hypertension, diabetes, and atherosclerosis [22]. Numerous studies have reported that heat treatment might bring about the so-called Maillard reaction, leading to liberation of free phenolic compounds due to thermal destruction of the cell and breakdown of the cell matrix, resulting in enhancement of measurable phenolic compounds and their antioxidant activities [23,24]. Previous studies demonstrated that total phenol content and antioxidant activity of water extracts of American ginseng [25] and ginseng [26] both increased significantly with heating temperature and time. Furthermore, Hwang et al. [21] reported that the total phenolic content and DPPH•-scavenging activities of hydroponically cultured ginseng leaves increased significantly after heating. In the present study, we first proved that total phenols, total flavonoids and the DPPH and ABTS antioxidant activities of GSEs increased with extension of the steaming time, implying that GSEs steamed for longer periods might have greater bioactivity. In other words, higher temperature probably led to the releases of more soluble or free phenols and flavonoids from cell walls and their conjugation through the Maillard reaction to various supramolecular organic compounds, thus contributing to increased antioxidant capacity. The results also provided evidence that total phenols, and particularly total flavonoids, were the main contributors to the scavenging activity estimated by the DPPH and ABTS assays. Moreover, the data presented in Figure 1 shows, based on the similarity of the enhancement patterns upon steaming, that the scavenging activity of total phenols is better reflected by the DPPH assay (Figure 1A,C), and that of total flavonoids by the ABTS assay (Figure 1B,D).

### 2.2. HPLC Profiles and Contents of Individual Ginsenosides in GSEs

As shown in Figure 2, remarkable differences were observed among the HPLC profiles of GSEs0–6. Compared to that of unsteamed GSE (i.e., GSE0; curve colored in yellow at the bottom), the HPLC profiles of all steamed GSEs (i.e., GSEs2–6; curves colored in purple, green, and blue, respectively) differed markedly with increasing steaming time. The most prominent changes were as follows: both the species (number of peaks) and contents (areas of individual peaks) of less- polar ginsenosides at retention times between 115 and 140 min increased, while those of polar ginsenosides between 25 and 90 min decreased. Notably, in GSE6, several polar ginsenosides, such as Rg_1_, Re, Rb_2_, Rb_3_, F_1_ and Rd, disappeared completely or nearly completely, while two less-polar ginsenosides (Rg_3_ and Rg_5_) reached their maximum values. These results, for the first time, clearly demonstrate that steaming for a longer period can cause conversion of polar ginsenosides (Table 1) into less-polar ginsenosides (Table 2) in GSEs.

To further characterize the dynamic mechanisms involved in conversion of ginsenosides in GSEs during steaming, the contents of individual ginsenosides were quantitatively determined using pure standards of 18 ginsenosides as references (Figure 2; curve colored in red at the top). GSE0 contained the highest total content of polar ginsenosides due to its considerable Re, Rg_1_, Rd and S-Rg_2_ contents (Table 1), while less-polar ginsenosides such as S-Rg_3_, Rg_5_, Rh_2_, PPT, R-Rg_3_, and CK were not detected (Table 2). With extension of the steaming time, most polar ginsenosides decreased markedly, with Re, Rg_1_, and Rd becoming scarce and Rb_2_, F_1_, and Rb_3_ completely disappearing in GSE6, resulting in a total content of polar ginsenosides 4.6 times lower than that in GSE0 (Table 1). On the other hand, six representative less-polar ginsenosides, among which only F_4_ was detected in GSE0, were strongly enhanced (especially F_4_ and S-Rg_3_) with increased steaming time from 2 to 6 h, resulting in at least 75.8 times higher total content of less-polar ginsenosides in GSE6 than in GSE0 (Table 2).

Similar results have been reported regarding the effects of heating on the transformation of ginsenosides in ginseng roots [21,27,28]. Li et al. [18] further studied the transformation mechanisms of ginsenosides in ginseng flowers during steaming and baking with different times and temperatures. In the present study, with the aid of solid experimental data (Table 1 and Table 2), we clarified the transformation mechanisms occurred in ginseng shoots for the first time. Among typical protopanaxadiols, Rd decreased by approximately 7.3 times in GSE6 compared to GSE0, while Rb_2_ and Rb_3_ both completely disappeared in GSE6 (Table 1). These results show that, as illustrated in Figure 3A, Rb_2_ and Rb_3_ were converted into Rd through hydrolysis of the Ara(p)- and Xyl-residues at C-20, and Rd was further transformed into F2 or S-Rg_3_ through loss of one Glc-residue at C-3 or C-20, respectively. If another Glc-residue at C-20 of F_2_ or S-Rg_3_ is hydrolyzed, CK or Rh_2_ is produced, respectively. CK and Rh_2_ were absent in GSE0, and gradually reached their highest levels in GSE4 and GSE6, respectively, during the steaming process (Table 2). Moreover, S-Rg_3_ could also be isomerized to R-Rg_3_, and R-Rg_3_ could be dehydrated at C-20 to yield Rg_5_ (Figure 3A); this series of transformations led to considerable amounts of Rg_5_ and Rg_3_ (both S- and R-forms) in GSE6 (Table 2).

As illustrated in Figure 3B, the protopanaxatriol Re was most likely transformed into S-Rh1 through hydrolysis of one Ara(p)-residue at C-20 or one Glc-residue at C-6, or to S-Rg_2_ through loss of one Glc-residue at C-20. Further hydrolysis of one Glc-residue at C-6 of S-Rh1 or one H_2_O at C-6 of S-Rg_2_ produced PPT and F4, respectively. As shown in Table 1, Re nearly disappeared and the content of S-Rh1 in GSE6 was 3.4 times higher compared to GSE0, while S-Rg_2_ was 1.4 times lower than that in GSE0, demonstrating conversion of a considerable amount of S-Rg_2_ and Re into F4 and S-Rh1, respectively, in GSE6 (Table 2). Although hydrolysis of one Glc-residue at C-6 of S-Rh1 could produce PPT, the content of S-Rh1 increased, implying that the conversion speed of S-Rh1 into PPT might be slower than that of Re into S-Rh1.

Together, for the first time, we determined the composition and contents of ginsenosides, and examined the contents of phenolic compounds and antioxidant activities in GSEs. The above results indicate that longer steaming time of GSEs not only increased their contents of both total phenols and total flavonoids, and elevated their DPPH and ABTS antioxidant capacities, but also enhanced the conversion from polar into less-polar ginsenosides.

### 2.3. Effects of GSEs on LPS-Induced RAW264.7 Cell Viability and Production of Three Inflammation Factors

As an in *vitro* system, RAW264.7 cells have been widely used to investigate inflammation mechanisms. Relative cell viability was first detected using the MTT assay to evaluate the cytotoxic effects of the four GSEs on RAW264.7 macrophages. As illustrated in Figure 4A, RAW264.7 cell viability decreased slightly and was maintained at 92.92% after treatment solely with LPS, and the GSE0–6 treatments did not significantly change the relative activity of cells (*p* ≥ 0.05), demonstrating that all four GSEs at all three concentrations (25, 50, or 100 µg mL^−1^) could be used in subsequent experiments with no cytotoxic effects on LPS-induced RAW264.7 cells.

Concentrations of NO in the cell culture media were measured to evaluate effects of GSEs on NO production by LPS-induced RAW264.7 cells (Figure 4B). LPS significantly stimulated NO levels compared to those of untreated cells. Further addition of GSEs0–6 to media at 25, 50, and 100 µg mL^−1^ significantly reduced NO release in a dose-dependent manner, and the extent of reduction at a particular dose was significantly enhanced with steaming time. NO is an essential inflammatory mediator of the host innate immune system, is involved in multiple biological processes, and participates in the inflammatory response to a variety of pathogens [29]. However, overproduction of NO causes oxidative damage to membrane lipids, DNA, proteins, and lipoproteins, triggering several disadvantageous cellular responses that are closely correlated to the pathophysiology of a variety of diseases and inflammatory disorders [30]. Therefore, inhibition of NO production is one possible method of screening anti-inflammatory drugs. Moreover, several reports have demonstrated that numerous polyphenols and flavonoids found in plants possess the ability to inhibit NO production in LPS-induced RAW264.7 cells [31,32,33]. In the present study, GSE4 and GSE6 clearly had stronger inhibitory effects on NO release than GSE0 and GSE2, indicating that enhancement of total phenols and flavonoids, and transformation of polar into less-polar ginsenosides, both occurred with extension of the steaming time, increasing the anti-inflammatory potential of GSEs.

To further explore the anti-inflammatory abilities of the GSEs, the concentrations of two inflammation cytokines in cell culture media were examined using an ELISA assay (Figure 4C,D). Similar to NO release, LPS significantly stimulated IL-6 and TNF-α levels compared to those of untreated cells. Further addition of the four GSEs to media, at 25, 50, and 100 µg mL^−1^, suppressed IL-6 and TNF-α levels in a dose-dependent manner. For IL-6 (Figure 4C), no significant differences were found between GSE0 and GSE2 at concentrations of 25 and 50 µg mL^−1^, while GSE2 exhibited superior inhibitory capacity compared to GSE0 at 100 µg mL^−1^. Compared to GSE0 and GSE2, GSE4 and GSE6 had stronger inhibitory effects at all three concentrations, with no significant differences found between the latter two treatments at the same concentrations. As shown in Figure 4D, a similar trend to that of IL-6 was observed in the inhibition of TNF-α levels by the four GSEs, but with no significant differences between 25 and 50 µg mL^−1^ GSE0, or between GSE0 and GSE2 at 25 µg mL^−1^.

IL-6 and TNF-α are two major pro-inflammatory cytokines responsible for immune response activation, and their overexpression is associated with various chronic diseases, including cancer, type II diabetes, and rheumatoid arthritis [34,35]. The pleiotropic activity of IL-6 not only contributes to cancer-related inflammation, but also plays crucial roles in DNA damage repair, the anti-oxidant defense system, proliferation, invasion, metastasis, angiogenesis, acute-phase protein synthesis in liver and lymphocyte differentiation, and metabolic remodeling [34,36,37]. TNF-α is secreted mainly by macrophages and lymphocytes in response to cell damage caused by infection or malignant transformation [35]. Thus, reducing levels of IL-6 and TNF-α is of great importance. Taken together, we firstly compared the anti-inflammatory effects of un-steamed and steamed GSEs by ELISA assays. These results show that steamed GSEs possesses strong anti-inflammatory ability, which was enhanced with increased steaming time.

In the present study, compared to that of LPS group, the levels of NO release, IL-6, and TNF-α decreased by 68.85% and 47.43%, 53.67% and 21.12%, and 49.75% and 29.53% with further additions of GSE6 and GSE0 at 100 µg mL^−1^, respectively. All in all, it is obvious that steamed ginseng shoot exerted even higher anti-inflammatory ability.

### 2.4. Effects of GSEs on Relative mRNA Expression of Four Inflammation Factors in LPS-Induced RAW264.7 Cells

To confirm whether the anti-inflammatory activities of the GSEs presented in Figure 4 resulted from regulation of expression of related genes, we finally determined the transcriptional expression levels of several inflammatory factors, including COX-2, iNOS, TNF-α and IL-6, in LPS-induced RAW264.7 cells treated with the four GSEs (GSEs0–6) at the three concentrations noted above (i.e., 25, 50, and 100 µg mL^−1^) using qRT-PCR (Table 3). LPS significantly up-regulated the mRNA expression of all four inflammatory factors (Figure 5). At 25 µg mL^−1^, GSE0 significantly enhanced iNOS mRNA expression, but GSE2-6 did not (Figure 5A). GSE0 and GSE2 did not significantly change the mRNA expression of IL-6 and TNF-α (Figure 5B,C) at the concentration of 25 µg mL^−1^, whereas the four GSEs significantly inhibited LPS-induced COX-2 mRNA expression at all three concentrations (Figure 5D). Finally, at all three concentrations of the four GSEs, the relative mRNA expression of all four inflammatory factors exhibited a dose-dependent reduction from 25 to 100 µg mL^−1^. In addition, the results are consistent with those presented in Figure 4C,D, suggesting that GSE4 and GSE6 exerted stronger inhibitory effects on both inflammatory mediators and inflammatory cytokines.

Inflammation is a crucial host defense response to invasion by pathogens, tissue damage, or other environmental assaults. Achieving balance between injury and restoration during the inflammatory response is essential to resolve the threat while ensuring minimal damage. If this balance is not achieved, an excessive or unregulated pro-inflammatory response might occur, leading to chronic tissue damage [38]. In the current study, the potential of steamed GSEs to decrease LPS-induced mRNA expression of four cytokines (iNOS, TNF-α, IL-6, and COX-2) was tested due to their significance in inflammatory conditions. As a prototypical inflammatory stimulus, LPS is able to rapidly activate the expression of genes responsible for high-throughput synthesis of inflammatory mediators in RAW264.7 cells. Overexpression of the iNOS gene can be induced in response to overproduction of various pro-inflammatory cytokines, including INF-γ, TNF-α, and IL-6, and is involved in several inflammatory responses including the nuclear factor-kappa B (NF-κB) pathway [39]. Most importantly, in macrophages stimulated with LPS, high levels of NO are generated mainly by iNOS, which is closely associated with inflammatory diseases such as atherosclerosis, septic shock, transplant rejection, and neurodegeneration [40,41]. In our study, steamed GSEs could inhibit the overproduction of NO (Figure 4B), which might result from inhibition of iNOS mRNA expression (Figure 5A). Apart from NO, TNF-α and IL-6, which are major pro-inflammatory cytokines in various immune cells including macrophages, monocytes and T-cells, also participate in the NF-κB signaling pathway [39]. Excessive production of TNF-α and IL-6 could lead to cell necrosis, tissue injury and degeneration, thus aggravating inflammation [42]. PGE2 is an inflammatory mediator generated at inflammatory sites by COX-2, which is an important enzyme in inflammatory pathogenesis that contributes to the development of many chronic inflammatory diseases, such as cardiovascular disease, cancer, and rheumatoid arthritis [43]. All of these inflammatory mediators and cytokines are closely related, participating in various signaling pathways in response to diverse inflammatory conditions.

Several studies have been carried out to explore the anti-inflammatory effects of ginseng root extract. Rhule et al. [44] found that Panax notoginseng extract could inhibit the production of TNF-α and IL-6 in a concentration-dependent manner, and could also attenuate the mRNA expression of COX-2 and IL-1β. Similarly, Jung et al. [39] reported that flower extract of Panax notoginseng attenuates the LPS-induced inflammatory response by blocking the NF-κB signaling pathway in murine macrophages. Wood-cultivated ginseng extract dose-dependently suppressed NO and PGE2, attenuated overexpression of iNOS and COX-2, blocked expression of TNF-α and IL-1β, and inhibited NF-κB activation in LPS-stimulated RAW264.7 cells [45]. In addition, Baek et al. [46] reported that the saponin fraction from red ginseng possessed greater ability to inhibit NO production and mRNA expression of inflammatory factor genes, such as iNOS, COX-2, TNF-α, and INF-β, than the non-saponin fraction. In the present study, for the first time, we systematically and comprehensively compared the anti-inflammatory effects of steamed and un-steamed GESs in molecular level along with identification of the composition and determination of the content of ginsenosides of the extracts. Compared to unsteamed ginseng shoot extract (GSE0), steamed ginseng shoot extract (GSE2-6) exhibited greater inhibition of the overproduction of inflammatory mediators (NO) and cytokines (TNF-α and IL-6; Figure 4) and of the corresponding mRNA expression of inflammatory factors (iNOS, TNF-α, and IL-6; Figure 5A–C). Taken together, the findings described above demonstrate that steamed GSEs have high potential as a potent anti-inflammatory agent.

### 2.5. Correlation Analysis

To determine the contributions of each GSE characteristic to the inflammatory response, correlation analysis was conducted among total polar ginsenosides, total less-polar ginsenosides, total phenols, total flavonoids, DPPH and ABTS antioxidant capacities, and various inflammatory factors [i.e., the release of NO, IL-6, and TNF-α (hereafter NO-r, IL-6-r, and TNF-α-r, respectively) and the relative mRNA expression levels of iNOS, IL-6, TNF-α, and COX-2 (iNOS-mR, IL-6-mR, TNF-α-mR, and COX-2-mR)] of GSEs at the three concentrations listed in Table 4. Total polar ginsenosides had a significant negative correlation with total less-polar ginsenosides, underlining the fact that less-polar ginsenosides were formed through conversion from polar ginsenosides (Figure 2). Total phenols exhibited a significant positive correlation with total flavonoids, showing that flavonoids accounted for a large proportion of phenolic compounds. Notably, both total phenols and total flavonoids were significantly positively correlated with ABTS antioxidant activity, but not with that of DPPH. In contrast, less-polar ginsenoside content was significantly positively correlated with DPPH antioxidant activity, but not with that of ABTS, while polar ginsenosides showed no such correlation, leaving many worthwhile topics for future studies.

Total polar ginsenosides were positively correlated at the level of *p* < 0.05 with TNF-α-r and TNF-α-mR at 25 µg/mL of GSEs; with IL-6-r, TNF-α-r, and COX-2-mR at 50 µg/mL; and with IL-6-r and iNOS-mR at 100 µg/mL; and at the level of *p* < 0.01 with IL-6-mR at all three concentrations, and with NO-r and TNF-α-r at 100 µg/mL. On the contrary, total less-polar ginsenoside content was negatively correlated at the level of *p* < 0.05 with NO-r, iNOS-mR, and TNF-α-r at 100 µg/mL; with IL-6-mR at both 50 and 100 µg/mL; and with IL-6-r at all three concentrations; and at the level of *p* < 0.01 with TNF-α-r at both 25 and 50 µg/mL, and with IL-6-mR at 25 µg/mL. These opposing (positive/negative) correlations implied that the less-polar ginsenosides formed through transformation of polar ginsenosides during the thermal process were the major contributors to inhibition of the overproduction of inflammatory factors.

Both total phenols and total flavonoids showed negative correlations with various inflammatory factors. Total phenols correlated at the level of *p* < 0.05 with TNF-α-mR at 25 µg/mL, with TNF-α-r and COX-2-mR at 50 µg/mL, and with IL-6-r and iNOS-mR at 100 µg/mL; and at the level of *p* < 0.01 with NO-r and iNOS-mR at 25 µg/mL, with COX-2-mR at 100 µg/mL and with TNF-α-mR at both 50 and 100 µg/mL. Meanwhile, total flavonoids correlated at the level of *p* < 0.05 with iNOS-mR at 25 µg/mL, COX-mR at 50 µg/mL, and TNF-α-mR at all three concentrations; and at the level of *p* < 0.01 with NO-r at 25 µg/mL and COX-2-mR at 100 µg/mL. These negative correlations suggest that the inhibitory effect of phenolic compounds on overproduction of inflammatory factors should not be neglected, although the correlation frequency and amplitude were not as great as those of ginsenosides (especially less-polar ginsenosides).

The antioxidant activities of DPPH and ABTS also exhibited negative correlations with the above inflammatory factors. DPPH was correlated at the level of *p* < 0.05 with TNF-α-r and TNF-α-mR, at both 25 and 50 µg/mL, and with IL-6-r and iNOS-mR at 100 µg/mL; and at the level of *p* < 0.01 only with IL-6-r at 25 µg/mL. ABTS was correlated at the level of *p* < 0.05 with NO-r and iNOS-mR at 25 µg/mL, and with TNF-α-mR at both 50 and 100 µg/mL; and at the level of *p* < 0.01 only with COX-2-mR at 100 µg/mL. These results implied that inhibiting the overproduction of the various inflammatory factors might also occur through the antioxidant activities of phenolic compounds, as well as those of ginsenosides, as indicated by the positive correlations described above. Interestingly, TNF-α-r and TNF-α-mR showed stronger correlations with the two antioxidant activity measures compared to other inflammatory factors.

In short, the inhibitory ability of GSEs on the overproduction of inflammatory factors might be due to all of the factors noted above. NO production was likely more closely linked with total phenols, total flavonoids and ABTS antioxidant activity at 25 µg/mL, whereas IL-6 release and mRNA expression were likely more closely related to the less-polar ginsenosides. Relative mRNA expression levels of various inflammatory factors were closely linked with different GSE characteristics, as manifested at different concentrations of GSEs. Notably, these inflammatory factors do not exist in isolation, but are interconnected and influence each other. These connections might result from the various inflammatory factors activating different inflammatory pathways, which could in turn activate the production of other inflammatory factors.

## 3. Materials and Methods

### 3.1. Chemicals and Reagents, Preparation of Plant Extracts, and Culture of RAW264.7 Cells

The LPS from *Escherichia coli* (0111: B4), Griess reagent, and dimethyl sulfoxide (DMSO) were purchased from Sigma-Aldrich Chemical (St. Louis, MO, USA). 3-(4,5-Dimethylthiazol-2-yl)-2,5-diphenyltetrazolium bromide (MTT), phosphate-buffered saline (PBS), Dulbecco’s modified Eagle’s medium (DMEM), fetal bovine serum (FBS), nonessential amino acids, and IL-6 and TNF-α ELISA kits were provided by Beijing BioDee Biotechnology Co. Ltd. (Beijing, China). Pure standards of the 18 ginsenosides and high-performance liquid chromatography (HPLC)-grade acetonitrile and methanol were purchased as described by Li et al. [18] and the chemicals and reagents for measuring DPPH and ABTS antioxidant capacity were prepared as described by Yang et al. [22]. Other reagents (analytical grade) were bought from Sinopharm Chemical Reagent Co. Ltd. (Beijing, China). Ultrapure water was prepared using a Milli-Q50 SP Reagent Water System (Millipore Corporation, Billerica, MA, USA).

Shoots of 6-year-old ginseng were purchased during the harvest season of field ginseng from Fusong City (Jilin, China), and then air-dried overnight at 60 °C in an oven. Ground ginseng shoot powder (10.0000 g) was placed in a conical flask and steamed in an autoclave at 120 °C for 0, 2, 4, or 6 h. After steaming treatment, the un-steamed (0 h) and steamed ginseng shoot powders were ultrasonically extracted (Power 500W, frequency 40 kHz; KQ-300DE NC ultrasonic cleaner, Kunshan ultrasonic instrument Co., Ltd., Jiangsu, China) with 100 mL of 70% aqueous ethanol at 30 °C for 30 min, and the mixture was allowed to settle to obtain the supernatant. The residue was subjected to ultrasonic extraction twice more with the same volume of solvent; all three supernatants were combined, then filtered with No. 1 filter paper (Whatman, Maidstone, UK) to remove residues, and finally filtered with a syringe filter (Tianjin Jinteng Experiment Equipment Co., Ltd., Tianjin, China). The filtrate was rotary-evaporated and dried in a water bath at 40 °C to prepare the crude extract. The yields of crude extracts from un-steamed ginseng shoots (GSE0) and ginseng shoots steamed for 2 h (GSE2), 4 h (GSE4), and 6 h (GSE6) were 26.44%, 23.36%, 22.98%, and 28.08%, respectively. The extracts, i.e., GSE0, GSE2, GSE4, and GSE6, were stored at −20 °C until use.

RAW264.7 mouse macrophages (RAW264.7 cells), purchased from the Cell Bank of the Chinese Academy of Medical Sciences (Beijing, China), were cultured in DMEM supplemented with 10% FBS, two antibiotics (penicillin, 100 units per mL; streptomycin, 100 µg mL^−1^; Solarbio, Beijing, China) and 0.5% non-essential amino-acids under a humidified atmosphere at 37 °C with 5% CO_2_ prior to use.

### 3.2. Evaluation of Phenolic Content and Antioxidant Capacity and HPLC Analyses of Ginsenosides

Total phenols were measured according to our modification of the Folin–Ciocalteu method [47]. In brief, Folin–Ciocalteu solution was added to a 96-well plate and sample solutions were added to individual wells. Then, Na_2_CO_3_ solution was added to each well and the plate was shaken in an orbital shaker. The microplate was incubated until absorbance measurement at 765 nm. Results are expressed as mg gallic acid equivalent (GAE)/100 g dry weight (d.w.) GSE. Total flavonoids were estimated using a modified aluminum chloride colorimetric assay [47]. Briefly, sample solutions were added to the wells of a 96-well microplate, followed by addition of NaNO_2_ to each well. After mixture, AlCl_3_ and NaOH were successively added before reading the absorbance. Results are expressed as mg rutin equivalent (RTE)/100 g d.w. GSE.

DPPH• scavenging activity was determined with our modified method [48]. In brief, samples were added to each well of a 96-well microplate, followed by addition of freshly prepared DPPH solution to each well. Subsequently, methanol was added to each well, and then the plate was shaken in an orbital shaker. After incubation, the absorbance at 517 nm was recorded. A standard calibration curve of Trolox (0–400 mg/L, *R*^2^ = 0.999) was plotted, and the results are expressed as µM Trolox equivalent (TE)/100 g d.w. GSE. The ABTS•^+^ scavenging capacities of samples were determined using our modified method [49]. Briefly, samples were added to individual wells in a 96-well microplate, followed by addition of ABTS working solution to each well. Absorbance at 734 nm was measured after incubation. A standard calibration curve of Trolox (0–800 mg/L, *R*^2^ = 0.991) was plotted, and the results are expressed as μmol TE/100 g d.w. GSE.

GSE0, GSE2, GSE4, or GSE6, prepared as described above, was dissolved in 2 mL HPLC grade methanol at a final concentration of 10 mg/mL for HPLC analysis, as reported in detail by Li et al. [18]. To quantitatively determine the ginsenoside levels in GSE0, GSE2, GSE4 and GSE6, a mixed solution of the 18 pure standards of ginsenosides was prepared and injected in six volumes (2, 4, 6, 10, 16, 20 µL) for linearity assessment. Linearity was established within the range of 0.88–10.8 µg, which exhibited good linearity (*R*^2^ > 0.999) with high precision, stability and repeatability and relative standard deviations of ≤5% for each of the 18 pure ginsenosides (Appendix A).

### 3.3. Assays of Cell Viability, and Release of NO and the Cytokines IL-6 and TNF-α

The MTT assay was first conducted to determine cell viability using the method reported by Xue et al. [50] with slight modifications. In brief, cultured cells were seeded into a 96-well plate at a density of 5 × 10^3^ cells per well and grown for 24 h. Then, the medium was removed and various concentrations of GSE0, GSE2, GSE4, and GSE6 were added for 2 h before treatment with LPS at a final concentration of 100 ng mL^−1^. After incubation for 24 h, 20 µL MTT (5 mg mL^−1^) was added, which was converted by the cells into visible formazan crystals after 4 h of incubation. The formazan crystals were then dissolved in 150 µL DMSO and the absorbance was measured at 570 nm using the microplate reader. Relative cell viability was calculated and compared to an untreated control.

NO release by the RAW264.7 cells (3 × 10^5^ cells per well) was assessed through measurement of the amount of sodium nitrite in the culture medium using the Griess test. Briefly, after 2 h of treatment with various concentrations of GSE0, GSE2, GSE4, and GSE6, cells were incubated for 24 h with LPS at a final concentration of 100 ng mL^−1^. Then, 50 µL Griess A (1% sulfanilamide in 5% H_3_PO_4_) and 50 µL Griess B (0.1% N-1-naphthyl-ethylenediamine-HCl) were added to 100 µL culture medium. Absorbance was detected at 540 nm, and a standard curve was plotted using NaNO_2_ to calculate the sodium nitrite concentrations.

The RAW264.7 cells were incubated for 24 h at a density of 1 × 10^6^ cells per well, then pretreated with various concentrations of GSE0, GSE, GSE4, and GSE6 (25, 50, and 100 µg mL^−1^) for 2 h. Subsequently, LPS (100 ng mL^−1^) was added and the supernatants were collected 24 h later. IL-6 and TNF-α release were detected using specific ELISA kits (Beijing BioDee Biotechnology Co. Ltd., Beijing, China) according to the manufacturer’s instructions.

### 3.4. qRT-PCR Assays of Inducible Nitric Oxide Synthase (iNOS), TNF-α, IL-6 and Cyclooxygenase-2 (COX-2) mRNAs

For qRT-PCR assays, RAW264.7 cells were seeded onto 6-well plates at 1 × 10^6^ cells per well and incubated for 24 h prior to treatment. Then, the supernatant was removed and after 2 h of treatment with different concentrations of GSE0, GSE2, GSE4, and GSE6, and the cells were incubated for 24 h with LPS at a final concentration of 100 ng mL^−1^. Next, the cells were washed with cold PBS and total RNA was extracted with TRIzol reagent (Tiangen Biotech, Beijing, China). Isolated RNA (1.5 µg) was converted to cDNA in a 20-µL reaction volume using a TIAN Script RT Kit (Tiangen Biotech) according to the manufacturer’s instructions. The cDNAs encoding iNOS, TNF-α, IL-6 and COX-2 mRNAs were then quantified by qRT-PCR. Briefly, all reactions were performed in 96-well plates with the following procedures: pre-denaturation at 95 °C for 10 min, 45 cycles of denaturation at 95 °C for 10 s, and annealing and extension at 60 °C for 30 s. GraPDH was used as the internal reference. Sequences of the specific primers used are listed in Table 2. Analyses of the data were performed using the 2^−∆∆Ct^ method with GraPDH for normalization of the samples.

### 3.5. Statistical Analysis

All experiments were conducted in triplicate, results are expressed as mean ± SD, and the statistical significance (two-sample equal variance-test, two-tailed distribution) was tested using SPSS software (ver. 20.0; SPSS Inc., Chicago, IL, USA) and Excel 2016 (Microsoft Corp., Redmond, WA, USA). Different letters mean significances at *p* < 0.05 based on Duncan’s multiple range test.

## 4. Conclusions

Ginseng shoots allow multiple harvests and have high biomass, but are not as widely used as ginseng root, which is only harvested once despite containing similar pharmacological and nutritional ingredients, such as ginsenosides, polysaccharides, triterpenoids, flavonoids, volatile oils, peptides, amino acids and fatty acids. To encourage better utilization of the massive ginseng shoot resources available, total phenols, flavonoids, and both polar and less-polar ginsenosides were examined, along with their antioxidant capacities and inhibitory effects on various inflammatory factors. Based on our results, when GSEs were steamed, less-polar ginsenosides, total phenols, total flavonoids, and their antioxidant capacities and anti-inflammatory effects all increased with steaming time. A longer steaming period might have two main consequences: release of more free and soluble phenolic compounds from the cell matrix after extensive heating, and transformation of certain polar ginsenosides into less-polar ginsenosides through the loss of sugar residues at various carbon positions during steaming, causing GSE4 and GSE6 to exhibit greater antioxidant and anti-inflammatory abilities. Further systematic and comprehensive studies should be conducted to explore which specific substances are the most important contributors to the anti-inflammatory effects of ginseng shoots. Nevertheless, our study provides a solid theoretical foundation for the anti-inflammatory effect of ginseng shoots and offers a valuable approach for producing GSE rich in less-polar ginsenosides and phenolic compounds.

## Figures and Tables

**Figure 1 ijms-20-02951-f001:**
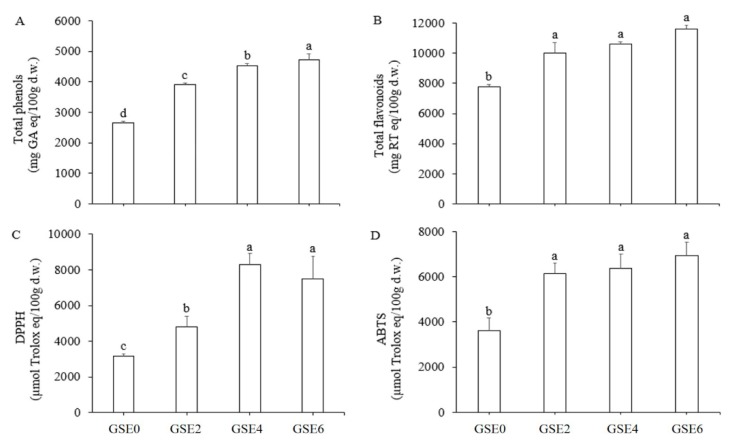
Changes in total phenols, total flavonoids, and the activities of two antioxidants in ginseng shoot extracts (GSEs) with steaming time. Gallic acid was used as the positive control for assays of total phenols (**A**), while rutin was used for total flavonoids (**B**). Trolox was used as the standard for 2,2′-diphenyl-1-picrylhydrazyl (DPPH•) (**C**) and 2,2′-azobis-(3-ethylbenzothiazoline-6-sulfonic acid) (ABTS•^+^) (**D**) scavenging activity assays. Absorbance was determined at 765 nm for total phenols, 410 nm for total flavonoids, 517 nm for DPPH, and 734 nm for ABTS. The results are presented as mean ± standard deviation (SD) of three independent experiments (*n* = 3). Total phenols and total flavonoids were expressed as µmol standards/100 g dry weight (d.w.), while DPPH• (**C**) and ABTS•^+^ (**D**) were expressed as µmol trolox/100 g d.w. Different letters indicate significant differences at *p* < 0.05 based on Duncan’s multiple range test.

**Figure 2 ijms-20-02951-f002:**
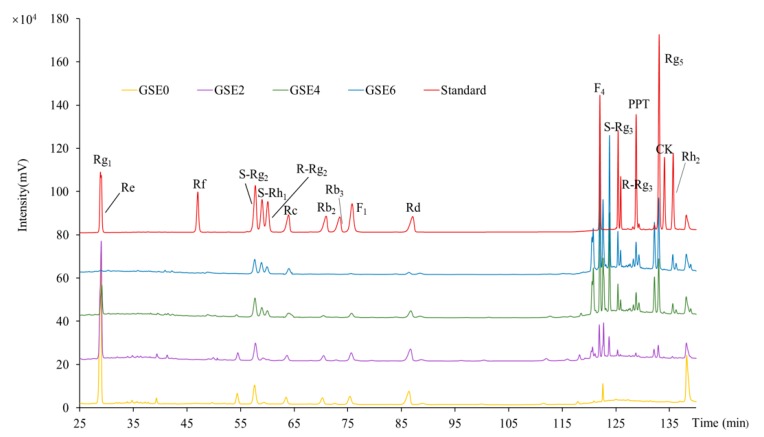
High-performance liquid chromatography (HPLC) profiles of 18 pure standards of ginsenosides and GSEs (ginseng shoots extracts) steamed for different times. High-performance liquid chromatography (HPLC) profiles of 18 pure standards of ginsenosides (red curve) and GSEs steamed for different times: unsteamed GSE (GSE0; yellow curve) and GSEs steamed at 120 °C for 2 h (GSE2; purple curve), 4 h (GSE4; green curve), and 6 h (GSE6; blue curve).

**Figure 3 ijms-20-02951-f003:**
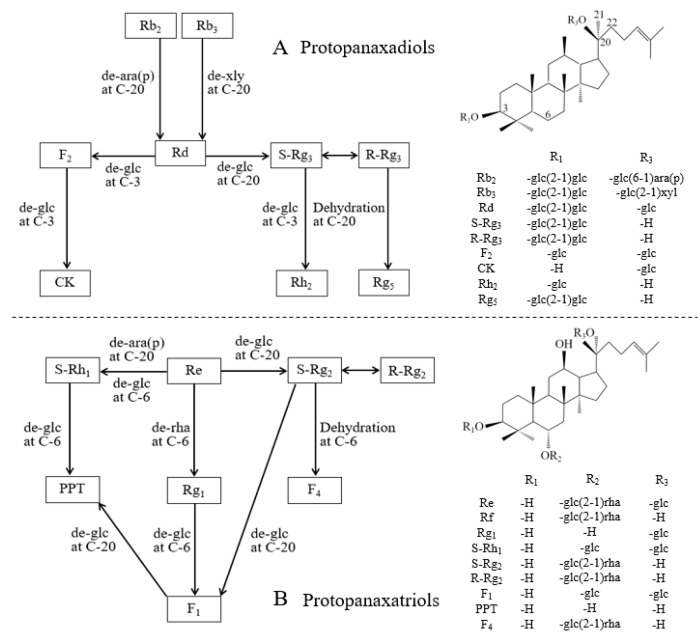
Chemical structures and possible transformation mechanisms of 18 representative ginsenosides in steamed GSEs. (**A**) Protopanaxadiols; and (**B**) protopanaxatriols. Glc, α-D-glucopyranosyl; Ara(f), α-L-arabinofuranosyl; Ara(p), α-L-arabinopyranosyl; Rha, α-L-rhamnopyranosyl; and Xyl, β-L-xylopyranosyl. Chemical links between C-20 and C-22 of F_4_ and Rg_5_ are double bonds.

**Figure 4 ijms-20-02951-f004:**
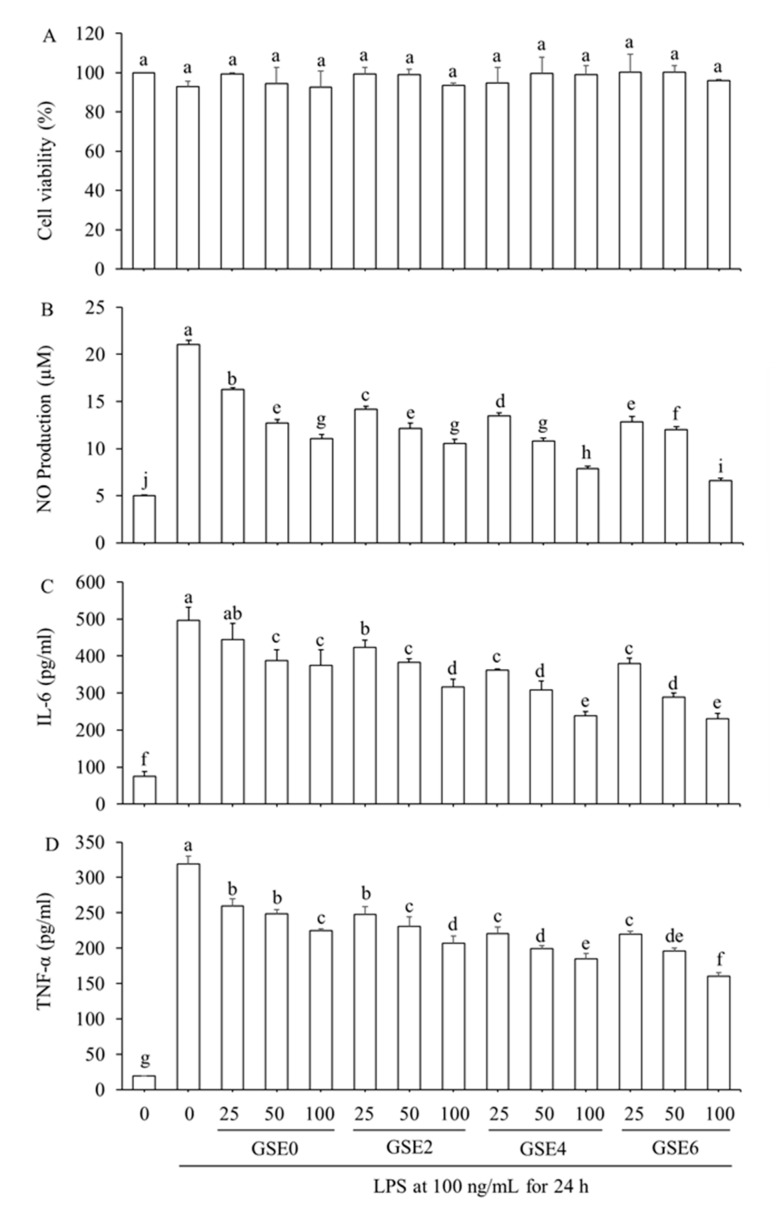
Effects of GSEs steamed for different times (0–6 h) on viability, nitric oxide (NO) production and the levels of two inflammatory cytokines in lipopolysaccharide (LPS)-induced RAW264.7 cells. (**A**) Relative cell viability; (**B**) NO production; (**C**) interleukin (IL)-6 level; and (**D**) tumor necrosis factor (TNF)-α level. RAW264.7 cells were pretreated with various concentrations (25, 50, or 100 µg mL^−1^) of GSE0, GSE2, GSE4, or GSE6 or vehicle for 2 h and then treated with 100 ng mL^−1^ LPS for 24 h. Results are representative of three independent experiments and are presented as means ± SD. Different letters indicate significant differences at *p* < 0.05 based on Duncan’s multiple range test.

**Figure 5 ijms-20-02951-f005:**
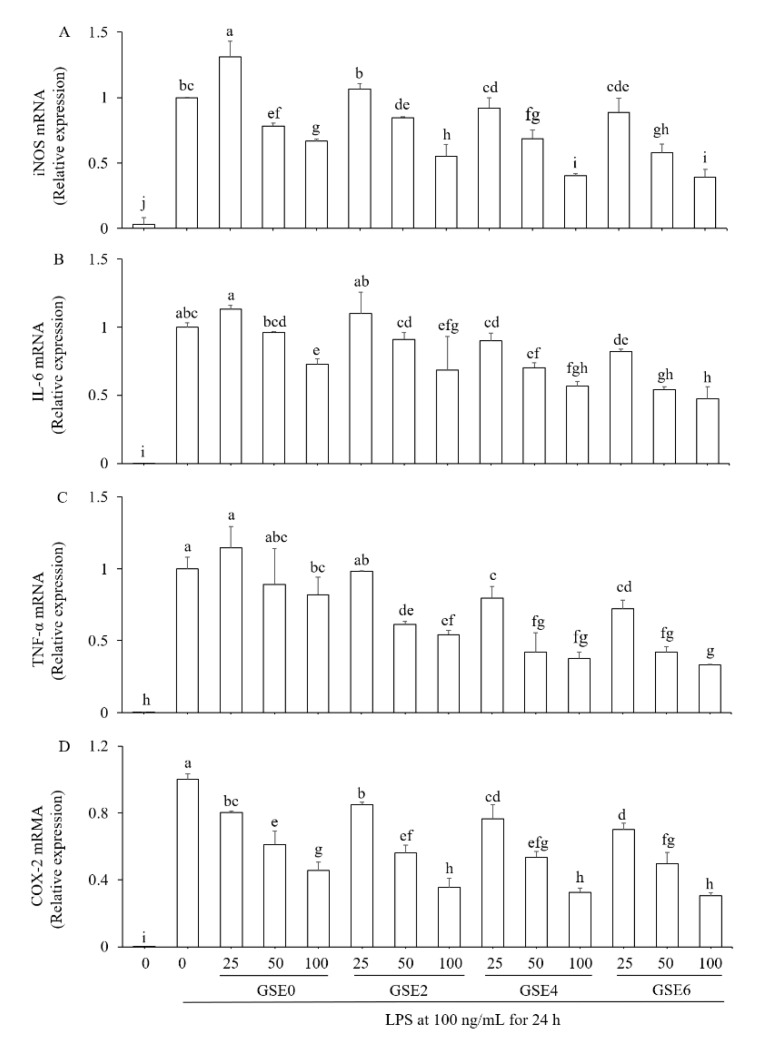
Inhibitory effects of GSEs steamed for different times (0–6 h) on relative mRNA expression of four pro-inflammatory factors in LPS-induced RAW264.7 cells. Cells were incubated with GSE0, GSE2, GSE4, GSE6 (25, 50, or 100 µg mL^−1^), or vehicle for 2 h and then treated with 100 ng mL^−1^ LPS at 37 °C. After a 24-h treatment with LPS, the mRNA expression levels of inducible nitric oxide synthase (iNOS) (**A**), IL-6 (**B**), TNF-α (**C**) and cyclooxygenase-2 (COX-2) (**D**) were determined using qRT-PCR and normalized to GraPDH levels. The results shown here are representative of three independent experiments and are presented as means ± SD. Different letters indicate significant differences at *p* < 0.05 based on Duncan’s multiple range test.

**Table 1 ijms-20-02951-t001:** Changes in contents of 11 polar ginsenosides in GSEs0–6 (mg/g).

Ginsenoside	GSE0	GSE2	GSE4	GSE6
Re	0.591 ± 0.031 ^a^	0.422 ± 0.023 ^b^	0.096 ± 0.008 ^c^	0.002 ± 0.000 ^d^
Rg_1_	0.241 ± 0.019 ^a^	0.218 ± 0.002 ^b^	0.059 ± 0.004 ^c^	0.003 ± 0.000 ^d^
Rd	0.154 ± 0.008 ^a^	0.146 ± 0.059 ^ab^	0.089 ± 0.012 ^b^	0.021 ± 0.002 ^c^
S-Rg_2_	0.125 ± 0.012 ^a^	0.115 ± 0.006 ^a^	0.121 ± 0.007 ^a^	0.088 ± 0.004 ^b^
Rb_2_	0.072 ± 0.004 ^a^	0.059 ± 0.000 ^b^	0.026 ± 0.000 ^c^	-
F_1_	0.061 ± 0.004 ^a^	0.058 ± 0.001 ^a^	0.031 ± 0.002 ^b^	-
Rc	0.058 ± 0.003 ^a^	0.046 ± 0.003 ^b^	0.052 ± 0.006 ^ab^	0.049 ± 0.003 ^b^
S-Rh_1_	0.022 ± 0.000 ^c^	0.021 ± 0.001 ^c^	0.062 ± 0.002 ^b^	0.075 ± 0.002 ^a^
Rb_3_	0.013 ± 0.001 ^a^	0.012 ± 0.002 ^a^	-	-
R-Rg_2_	- *	0.001 ± 0.000 ^c^	0.044 ± 0.007 ^b^	0.052 ± 0.002 ^a^
Rf	-	-	-	-
Total	1.338 ± 0.083 ^a^	1.099 ± 0.099 ^b^	0.578 ± 0.049 ^c^	0.290 ± 0.013 ^d^

Different letters within individual rows indicate significant differences at *p* < 0.05 based on Duncan’s multiple range test. Data are means ± SD (*n* = 3). The order of ginsenosides from top to bottom is based on the contents of individual ginsenosides in GSE0 from high to low. Among the ginsenosides listed, Rd, Rb_2_, Rc, and Rb_3_ are protopanaxadiols, while Re, Rg_1_, S-Rg_2_, F_1_, S-Rh_1_, R-Rg_2_, and Rf are protopanaxatriols. * not detected.

**Table 2 ijms-20-02951-t002:** Changes in contents of seven less-polar ginsenosides in GSEs0–6 (mg/g).

Ginsenoside	GSE0	GSE2	GSE4	GSE6
F_4_	0.006 ± 0.001 ^c^	0.046 ± 0.007 ^b^	0.167 ± 0.004 ^a^	0.170 ± 0.004 ^a^
S-Rg_3_	- *	0.025 ± 0.001 ^c^	0.120 ± 0.013 ^b^	0.149 ± 0.004 ^a^
Rg_5_	-	0.012 ± 0.001 ^b^	0.050 ± 0.002 ^a^	0.047 ± 0.004 ^a^
Rh_2_	-	0.006 ± 0.001 ^c^	0.031 ± 0.002 ^b^	0.046 ± 0.006 ^a^
PPT	-	0.003 ± 0.000 ^c^	0.015 ± 0.001 ^b^	0.024 ± 0.183 ^a^
R-Rg_3_	-	0.001 ± 0.000 ^c^	0.007 ± 0.000 ^b^	0.014 ± 0.002 ^a^
CK	-	0.004 ± 0.000 ^b^	0.010 ± 0.001 ^a^	0.004 ± 0.000 ^b^
Total	0.006 ± 0.000 ^b^	0.098 ± 0.011 ^b^	0.400 ± 0.024 ^a^	0.455 ± 0.203 ^a^

Different letters within individual rows indicate significant differences at *p* < 0.05 based on Duncan’s multiple range test. Data are means ± SD (*n* = 3). The order of ginsenosides from the top to bottom is based on the contents of individual ginsenosides in GSE6 from high to low. Among the ginsenosides listed, F_4_ and PPT are protopanaxadiols, while S-Rg_3_, Rg_5_, Rh_2_, R-Rg_3_, and CK are protopanaxatriols. * not detected.

**Table 3 ijms-20-02951-t003:** Primers used for qRT-PCR.

Gene	Primer Sequence (5′ to 3′)	bp
*iNOS* Forward	AATGGCAACATCAGGTCGGCCATCACT	27
*iNOS* Reverse	GCTGTGTGTCACAGAAGTCTCGAACTC	27
*COX-2*	TGAA GCCGTACACATCATTTGAA	23
*COX-2*	TGGTCTCCCCAAAGATAGCATCT	23
*IL-6*	GGGGATGTCTGTAGCTCATTCTGCTCTG	28
*IL-6*	AAGGACTCTGGCTTTGTCTTTCTTGTTA	28
*TNF-α*	CTGTGAAGGGAATGGGTGTT	20
*TNF-α*	CAGGGAAGAATCTGGAAAGGTC	22
*GraPDH*	TGTTTCCTCGTCCCGTAG	18
*GraPDH*	CAATCTCCACTTTGCCACT	19

**Table 4 ijms-20-02951-t004:** Correlations among total polar ginsenosides, phenolic compounds, total less-polar ginsenosides, antioxidant capacities, and inflammatory factors at various concentrations of GSEs.

	TPG	TLPG	TP	TF	DPPH	ABTS	NO-r	IL-6-r	TNF-α-r	iNOS-mR	IL-6-mR	TNF-α-mR	COX-2-mR
TPG	1												
TLPG	−0.988 *	1											
TP	−0.923	0.918	1										
TF	−0.922	0.891	0.984 *	1									
DPPH	−0.924	0.965 *	0.934	0.870	1								
ABTS	−0.842	0.817	0.976 *	0.982 *	0.837	1							
GSE0, GSE2, GSE4, GSE6 at 25 µg/mL													
NO-r	0.921	−0.903	−0.996 **	−0.996 **	−0.901	−0.985 *	1						
IL-6-r	0.898	−0.952 *	−0.895	−0.817	−0.995 **	−0.782	0.855	1					
TNF-α-r	0.974 *	−0.994 **	−0.941	−0.902	−0.987 *	−0.846	0.920	0.974 *	1				
iNOS-mR	0.930	−0.930	−0.999 **	−0.979 *	−0.946	−0.967 *	0.992 **	0.911	0.953 *	1			
IL-6-mR	0.993 **	−0.991 **	−0.880	−0.870	−0.922	−0.777	0.873	0.906	0.971 *	0.892	1		
TNF-α-mR	0.986 *	−0.981 *	−0.974 *	−0.962 *	−0.955 *	−0.911	0.968 *	0.925	0.984 *	0.979 *	0.965 *	1	
COX-2-mR	0.846	−0.824	−0.575	−0.599	−0.663	−0.440	0.580	0.659	0.760	0.592	0.887	0.745	1
GSE0, GSE2, GSE4, GSE6 at 50 µg/mL													
NO-r	0.625	−0.725	−0.736	−0.605	−0.876	−0.683	1						
IL-6-r	0.980 *	−0.989 *	−0.854	−0.831	−0.928	−0.734	0.661	1					
TNF-α-r	0.978 *	−0.992 **	−0.959 *	−0.927	−0.982 *	−0.876	0.769	0.962 *	1				
iNOS-mR	0.900	−0.883	−0.661	−0.677	−0.741	−0.531	0.384	0.938	0.817	1			
IL-6-mR	0.991 **	−0.969 *	−0.868	−0.881	−0.873	−0.780	0.532	0.978 *	0.944	0.944	1		
TNF-α-mR	0.920	−0.930	−0.995 **	−0.963 *	−0.961 *	−0.956 *	0.797	0.867	0.970 *	0.662	0.860	1	
COX-2-mR	0.966 *	−0.932	−0.968 *	−0.988 *	−0.881	−0.941	0.575	0.894	0.949	0.781	0.942	0.947	1
GSE0, GSE2, GSE4, GSE6 at 100 µg/mL													
NO-r	0.995 **	−0.987 *	−0.881	−0.877	−0.911	−0.782	1						
IL-6-r	0.971 *	−0.982 *	−0.974 *	−0.944	−0.980 *	−0.903	0.951 *	1					
TNF-α-r	0.991 **	−0.959 *	−0.922	−0.943	−0.877	−0.865	0.979 *	0.949	1				
iNOS-mR	0.965 *	−0.977 *	−0.978 *	−0.946	−0.981 *	−0.910	0.943	1.000 **	0.943	1			
IL-6-mR	0.996 **	−0.974 *	−0.893	−0.905	−0.887	−0.813	0.995 **	0.945	0.994 **	0.937	1		
TNF-α-mR	0.934	−0.932	−0.999 **	−0.981 *	−0.944	−0.968 *	0.895	0.982 *	0.929	0.985 *	0.904	1	
COX-2-mR	0.895	−0.881	−0.996 **	−0.990 **	−0.898	−0.992 **	0.845	0.950 *	0.904	0.955 *	0.865	0.992 **	1

** and * indicate highly significant (*p* > 0.01) and significant (*p* > 0.05) correlations, respectively. TPG, total polar ginsenosides; TLPG, total less-polar ginsenosides; TP, total phenols; TF, total flavonoids; DPPH and ABTS, DPPH and ABTS antioxidant activities in respective; NO-r, NO release; IL-6-r, IL-6 release; TNF-α-r, TNF-α release; iNOS-mR, iNOS mRNA; IL-6-mR, IL-6 mRNA; TNF-α-mR, TNF-α mRNA; COX-2-mR, COX-2 mRNA.

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
