# Peer review of "Phenolic Compounds and Ginsenosides in Ginseng Shoots and Their Antioxidant and Anti-Inflammatory Capacities in LPS-Induced RAW264.7 Mouse Macrophages"

_ijms, 2019, doi:10.3390/ijms20122951_

Round 1

Reviewer 1 Report

After revising the new version of the manuscript, all the changes/modifications/suggestions done by the referee have been successfully accomplished. That is why I recomment publication of the manuscript in its present form in the International Journal of Molecular Sciences.

Author Response

Response to Reviewer 1 Comments

Point 1: After revising the new version of the manuscript, all the changes/modifications/suggestions done by the referee have been successfully accomplished. That is why I recommend publication of the manuscript in its present form in the International Journal of Molecular Sciences.

Response 1: The manuscript in its present form was revised a great deal according to your kind advice. Many thanks to you for all the detailed suggestions, comments and corrections in helping us improving the manuscript.

Reviewer 2 Report

Line 66, it should be "no studies have" instead of "no study have". 

In terms of the supplementary tables, the authors should make the peak area names readable. 

Author Response

Response to Reviewer 2 Comments

Point 1: Line 66, it should be "no studies have" instead of "no study have".

Response 1: The manuscript in its present form was revised a great deal according to your kind advice. Many thanks to you for all the detailed suggestions, comments and corrections in helping us improving the manuscript. And thanks for pointing out a further problem, we have corrected the word "study" to "studies". (Line 66)

Point 2: In terms of the supplementary tables, the authors should make the peak area names readable.

Response 2: Thanks for your suggestions, we have added the name of each ginsenoside in the note to the supplementary Tables as follows: Rg1, ginsenoside-Rg1; Re, ginsenoside-Re, Rf, ginsenoside-Rf; S-Rg2, ginsenoside-Rg2(S-form); S-Rh1, ginsenoside-Rh1(S-form); R-Rg2, ginsenoside-Rg2(R-form); Rc, ginsenoside-Rc; Rb2, ginsenoside-Rb2; Rb3, ginsenoside-Rb3; F1, ginsenoside-F1; Rd, ginsenoside-Rd; F4, ginsenoside-F4; S-Rg3, ginsenoside-Rg3 (S-form); R-Rg3, ginsenoside-Rg3 (R-form); PPT, S-protopanaxatriol; Rg5, ginsenoside-Rg5; CK, ginsenoside-compound K; Rh2, ginsenoside-Rh2.

This manuscript is a resubmission of an earlier submission. The following is a list of the peer review reports and author responses from that submission.

Round 1

Reviewer 1 Report

The paper reports a study of a steaming-based extraction procedure of phenolic compounds and ginsenosides from ginseng shoots and the evaluation of the antioxidant and anti-inflammatory properties of the extracts. Different times of steaming are tested; the antioxidant capacity is measured with the DPPH and ABTS methods and the anti-inflammatory activity in lipopolyssacharide-induced RAW264.7 cells in terms of inhibition of mediators and cytokines, such as nitric oxide, interleukin-6 and tumor necrosis factor is studied as well. Besides, the relative mRNA expression levels of four inflammation factors is also calculated for the different steamed extracts. An important statistical study is accomplished in the paper.

The paper seems to be interesting. However, novelty of the manuscript is not clear or not well stated. The paper is too long and it is difficult to see where the authors’ work starts and where finishes, since many of the results reported have been previously done by other scientists and so stated by the authors. Besides, several details need to be corrected and much improvement needs to be done. That is why I recommend major revision of the paper prior to its publication in the International Journal of Molecular Sciences.

Specific remarks

1) In the title, the word ‘Phenolics’ is used, but it suggests that something is missing there. It seems authors use this word as a name, when actually it is used in general as an adjective. I would suggest to use better ‘Phenolic compounds’ or ‘Phenolic species’ instead of ‘Phenolics’ alone.

2) In the Introduction section (line 58), the expression ‘… we mainly focus on its root…’ is confusing, since it seems that authors focus their study on ginseng roots throughout the paper, instead of on ginseng shoots.

3) As suggested previously, the authors should stress better the advantages of the methodology proposed as well as the novelty (extremely important), since many of the assays reported have been previously reported in other published papers, as they state.

4) Section 2.1 (lines 77 and 79), the use of the word ‘significant’ seems to come from the application of an statistical test. This test should be well stated in the text correspondingly.

5) Line 95, can the word ‘supermolecular’ be changed to ‘supramolecular’?

6) In Figure 1, letter ‘a’ is repeated many times at the top of the columns of Figures B,C and D. Nothing is explained about that in text or even in the caption of the figure. The name of the statistical test applied, for example, should be specified clearly there. In table 1 is done, why not here? In the caption of this figure, the authors say that micromole standards/100 g dry weight is the unit for Y-axis, when, in fact, there are two types of units: for phenols and flavonoids and for trolox and ABTS. Please, check.

7) Suggestion for caption of Figure 2, line 137: Change: ‘High-performance liquid chromatography (HPLC) profiles of 18 pure standards of ginsenosides (red curve) and GSEs steamed for different times. High-performance liquid chromatography (HPLC) profiles of unsteamed GSE (GSE0; yellow curve) and GSEs steamed at 120 °C for 2 h (GSE2; purple curve), 4 h (GSE4; green curve) and 6 h (GSE6; blue curve) are shown.’ by ‘High-performance liquid chromatography (HPLC) profiles of 18 pure standards of ginsenosides (red curve) and GSEs steamed for different times: unsteamed GSE(GSE0; yellow curve) and GSEs steamed at 120 °C for 2 h (GSE2; purple curve), 4 h (GSE4; green curve) and 6 h (GSE6; blue curve)’.

8) Line 170, the number 2 in H2O should be in subindex format.

9) Section 2.3, at the end of this section, perhaps some percentage of variation of the data could be included for simplification and clarification purposes, as a conclusion of this part.

10) Figure 4. Which statistical test is used in this figure? In general, some information about the text applied should be included in the text and in the caption of figures and tables.

11) Section 2.5, line 334. I think that according to table 6, total polar ginsenosides are also correlated at the level of p<0.01 at 100 microg/ml with IL-6-mr. Please check this discussion section.

12) Materials and methods section. This section has much information missing: conditions of ultrasonic extraction and type of instrumentation used; instrumentation description in general, with company and country; description of parameters employed in each assay; different incubation conditions (time, temperature, mode, etc.). Precision, stability, repeatability and RSD values corresponding to the pure standards of ginsenosides could be included in a Supplementary Material section. Line 475, please check the value ‘32 s’ for annealing. Line 476: here, Table 3 should be instead of Table 2. In section 3.5, a whole description regarding statistical analysis (techniques) and variables or parameters employed should be done. Nothing is said about it in the text.

13) Revise format of reference 50.

Reviewer 2 Report

The novelty of this study was not sufficient. There have been plenty studies about the effects of processing conditions on gensinosides. This presented study did not show enough originality. 

In addition, some statistic analysis result in this manuscript seems incorrect. For instance, in Figure 4C, the second and the third bars had similar means and their standard deviations were not too small. It is hard to believe those two groups are statistically significant different. 

Some descriptions of the results were not consistent with what was shown in the figures. For example, in Line 256, the authors said "GSE0 and GSE2 did not significantly change the mRNA expression of IL-6 and TNF-α". It was not consistent with the results presented.